# The Impact of Campus-Based Therapy Dogs on the Mood and Affect of University Students

**DOI:** 10.3390/ijerph20064759

**Published:** 2023-03-08

**Authors:** Nicole Peel, Kathy Nguyen, Caterina Tannous

**Affiliations:** School of Health Sciences, Western Sydney University, Penrith, NSW 2751, Australia

**Keywords:** therapy dogs, Ottawa charter, university, mood affect

## Abstract

University students experience a high level of stress, which could potentially affect how they manage stressful situations beyond university, such as when entering the workforce. Although universities offer counseling services and various health promotion programs, there is reluctance and negative perceptions about utilizing these from students. Further research is needed to explore the effectiveness of therapy dog interventions in human interactions that is quantifiable and embraces the elements of health promotion. This study aimed to investigate the impact of therapy dog interventions on students’ moods across a multi-campus university during a 2-week final examination period. Two hundred and sixty-five students participated in the study involving a multi-campus university. The intervention group and control group completed a questionnaire involving the positive affect negative affect schedule (PANAS), a 20-item scale that measures a person’s affect at the time. The intervention group (n = 170) had a higher average of total PANAS scores (mean = 77.63, standard deviation = 10.975) compared to the control group (n = 95) (mean = 69.41, standard deviation = 13.442). The results were statistically significant (mean difference = 8.219, 95% CI = 5.213–11.224, *p* < 0.05) with a t-score of 5.385. Students who engaged with therapy dogs on campus during the examination period were more likely to have a more positive affect. The results suggest that universities should include therapy dog programs within their health promotion programs for students, as these may help improve their mood and reduce the stress associated with university examinations.

## 1. Introduction

### 1.1. Animals and Health

Animal-assisted therapy is a goal-oriented health intervention using a trained animal for a wide range of people to achieve specific therapeutic outcomes, such as reducing stress levels [1,2,3]. Any species of animal can possibly be part of the intervention [2,4]. However, the bulk of animal-assisted therapy studies and health benefits demonstrated have focused on dogs because they are easily trainable compared to other species [2,4]. Animal-assisted therapy has the goal of improving a person’s cognitive, emotional, social, and physical functioning [4]. Studies have found that relationships between humans and animals have shown benefits in physiological and psychological wellbeing [1]. Trained professionals can therefore implement a goal-directed intervention with a trained animal to achieve outcomes such as stress reduction [5].

The therapeutic use of animals has existed for decades, such as in clinical settings with mental health conditions for people with schizophrenia, and depression or with patients with drug or other substance addictions [2,4]. Other populations benefiting from animal-assisted therapy include patients with cancer, heart conditions, and neurological conditions [2,4]. Hospitalized children and elderly patients have also been seen to benefit from animal-assisted therapy [2,4]. The benefits of having a relationship with animals include improved communication and social behaviors, quality of life, and an internal locus of control [2,4]. Being with animals also benefits patients by creating a calming, happy and comfortable feeling [2,4].

In populations with mental health conditions, interacting with animals has been shown to produce significant decreases in depression, anxiety, pain, and pulse [6]. Animal-assisted therapy has also been shown to reduce anxiety in patients diagnosed with major depression [7]. Children with disabilities show improvements in their behavioral management, and animals can also provide comfort before medical procedures [8]. Hospital patients in animal-assisted therapy groups have significant decreases in depression and anxiety levels based on measurements of their blood pressure, pulse, and salivary cortisol levels [9]. In patients with post-traumatic stress disorder, there have been significant decreases in symptoms of depression and anxiety, emotional distress, and alcohol use [2]. Psychiatric inpatients were seen to have improvements in their communication and social behaviors [4]. Furthermore, there were significant improvements noted in the living profile skills and social contact scores of inpatients with schizophrenia in the therapy dog intervention group [2]. However, despite these positive results, existing research is largely description-based, using small samples of participants and rarely including a control group [3]. In studies that did use a control condition, it was often difficult to compare the effectiveness of animal-assisted therapy for specific conditions as different scales were used amongst the studies [5]. The generalizability of results is inconsistent due to the study format, study duration, and sample size [5].

### 1.2. Animals and University Students

University students have substantially low mental health functioning and experience stress at a high level [10,11]. Studying at university is linked to social and academic challenges that cause high rates of psychological stress [12,13]. Exams are an important aspect of university assessment; however, they can cause negative outcomes in the form of stress and anxiety, increased tension, depression, negative impacts on immune functioning, lower expectations, and lower confidence in grades and capabilities [14,15]. This can lead to poor academic performance and a negative view of their course and overall university experience [14,15]. 

Strategies to address the stress of university students are available with varying degrees of effectiveness and include cognitive behavior therapy and mindfulness activities [16]. These strategies are often underutilized, as students refrain from using services where they are required to disclose their problems [13]. Existing mainstream interventions for university students, such as mindfulness programs, are often limited by a lack of specific training, resources, time, and accessibility to large groups of students [15]. Thus, some students perceive existing supports as inaccessible or inconvenient and, as such, continue to experience anxiety regarding the examinations [1]. There is a need for preventative programs that address wellbeing and mental health that are widely available and can attract large numbers of students [16]. Interest has risen for therapy dog interventions at universities as these are cost-effective, easily accessible, require no training for students or university staff, and can reach many students who experience stress before exams [14,15]. Therapy dogs in a university environment, embedded as part of the institution’s student wellness program, are an exemplar of a health promotion initiative using the Ottawa Charter (1986). Specifically, the initiative demonstrates achievement of health and wellbeing through institutional policy that creates a supportive environment, develops students’ personal skills and connections to others, and reorients student support to broader initiatives that are inclusive of all students [17].

Evidence is growing for non-clinical population groups [11]. Animal-assisted therapy has shown stress-relieving effects in environments of high stress and anxiety [1], such as in residential health and educational contexts [9]. The use of therapy dogs for university students has shown significant reductions in outcome measures of psychological and physiological stress [18]. There is a substantially larger statistical effect size compared to standard interventions for addressing student wellbeing, such as counseling and mindfulness programs [19]. Students who lived away from home also reported less homesickness and increased satisfaction due to having the dogs on campus, which created a relaxed and soothing environment [19]. It was found that the use of dogs improved student perceptions of the accessibility of counseling services and was able to reach many students [20]. Generally, throughout various studies, therapy dogs were seen to be a source of comfort and stress relief that promoted a sense of belonging within the university community [9]. Although therapy dog interventions are seen as a suitable, cost-effective, and effective method for student mental health and wellbeing, only a few studies have reported their effectiveness with a standardized measure of stress [15,19].

Various studies have explored the impact of therapy dog interventions on students’ physiological stress factors, such as blood pressure [18], nerve growth factor, and alpha amylase [14]. Barker et al. [14] conducted a randomized exploratory study with students (n = 78) randomly assigned to a 15-min therapy dog intervention or a control condition group. It was concluded that lower scores were obtained for nerve growth factor and alpha amylase from saliva were found among the intervention group, indicating a reduction in physiological stress (intervention first, *p* = 0.0001, d = 1.87; intervention second, *p* = 0.0004, d = 1.63) [14] In another study, Wood et al. [14] ran unstructured group interventions with a guide dog and found that the intervention group had statistically significant results post-intervention based on a reduction in blood pressure [14]. The study by Jarolmen and Patel [1] with 86 students divided into the intervention or control group had their blood pressure readings measured using ReliOn at 15-min intervals. The intervention group was permitted to interact with the therapy dogs between the readings, while the control group only had their blood pressure measured [1]. It was found that there was a statistically significant decrease in both systolic and diastolic blood pressure for the experimental group and a significant decrease in blood pressure compared to the control group [1]. 

Several therapy dog studies have also found improvements in psychological factors such as stress. An intervention where participants (n = 132) had a duration of 20 min with therapy and dog handlers was found to improve mood measured on the Mood Adjective Checklist based on 20-min durations. When comparing a therapy dog and handler group to controls of therapy dog only and handler only groups, the mean difference was 2.62, 4.15, and −0.26 (*p* < 0.001), respectively [11]. Additionally, a decrease in anxiety was observed on the State Trait Anxiety Inventory; the mean difference was −13.73, −12.98, and −2.02 (*p* < 0.001) for the therapy dog and handler group, the therapy dog only group, and the handler only group, respectively [11]. Furthermore, interacting with a dog (n = 67) was found to reduce negative mood and increase positive mood on the Positive Affect Negative Affect Schedule compared to the control condition, with t score = −3.57 and t score = 3.52 for negative and positive affect, respectively [21]. 

In the study by Ward-Griffin et al. [22], students (n = 246) completed a questionnaire before and after the therapy dog session and reported improvements such as reducing negative affect, increasing perceived social support, and decreasing perceived stress compared with those in the delayed-treatment control group. Lannon and Harrison [9] completed a study where the university librarian asked participants to complete a questionnaire to rank their stress before and after interacting with the therapy dogs on a 0–4 scale from not stressed to extremely stressed. The undergraduate students (80.7%) reported a two-point drop in stress, while 38.6% reported a one-point drop, and 5.3% reported no change in stress [9]. Wood et al. [18] ran unstructured group interventions with a guide dog for students (n = 131) and found some clinically significant reductions based on the State Trait Anxiety Inventory with no comparison group. Additionally, it was found that therapy dog interventions had resulted in the lowering of stress visual analog scale (SVAS) scores (first *p* = 0.0001, d = 1.87; second *p* = 0.0004, d = 1.63) whereas there were no SVAS differences for the control group [14]. 

Studies also discussed students’ perceptions of the use of therapy dogs on a university campus. Some studies have reported that students responded positively to the use of therapy dogs within the university [15,21]. Barker et al. [15] conducted a study where students (n = 694) completed pre-and-post surveys in which participants included demographic data and comments indicating their perceived stress immediately before and after interacting with the therapy dogs in a large meeting room, a week before the final exam week. Of these participants, 92.9% reported less stress after the interaction, 80.5% reported providing positive comments about their experience, and 4.1% provided negative comments [15]. 

Although there are existing studies on the impact of therapy dog interventions on university students, there are limitations on the sample size and the use of a control group to compare the effects of the intervention. Limited studies found conclusive results by measuring stress levels associated with exams [15,18]. Therapy dog interventions are still used for the purpose of managing student stress as they are resource-efficient, can target a large number of students, and can capture the interest of students more than other mainstream interventions [1,15]. Due to the non-conclusive findings on the efficacy of the use of therapy dogs and their effectiveness in lowering student stress [15,18], there is a need to further investigate the impact that therapy dogs may have on a large sample of university students of varying contextual backgrounds with the use of an appropriate outcome measure that may add value to health promotional practices. Furthermore, it would be valuable to explore whether any independent factors influenced the effectiveness of the therapy dog interventions for the students. 

The purpose of this study was to investigate the impact of dog exposure on the mood and affect of student participants on a multi-campus university during the end of semester exams, with three sub-questions. 

(1)How does exposure to therapy dogs on a university campus impact students’ affect and mood during the final examination period compared to no exposure?(2)Were PANAS scores influenced by gender, age, year group, enrolment factors, and surveys on campus?(3)What were the students’ perspectives regarding having therapy dogs on campus and having the university fund the program through their Student Services Fees?

## 2. Materials and Methods

This study used a mixed method design incorporating qualitative and quantitative methodology. 

The quantitative component allows for reliability, validity, and generalizability of the results to the research questions [23]. This methodological approach examines the differences found in outcomes between groups and investigates the influences of other independent factors [24]. Thus, there is an exploration of causal relationships [24] and whether the use of therapy dogs impacts students’ mood and affect. The qualitative component examines the perceptions the students have of how dogs impact them and their perspective on the use of funding for such programs at the university. The use of the qualitative findings can help to bring understanding to the changes in mood and affect of university students.

### 2.1. Sampling and Recruitment 

Convenience sampling [25] was used in the recruitment of participants in the intervention and control groups who completed a survey. To address the limitations of convenience sampling, the study was held across four campuses of a public funded metropolitan university, in easily accessible areas of the campus and announced on the university’s Facebook page to increase awareness and participation.

### 2.2. Intervention Group

Therapy dogs from Delta Society Australia and PAWS therapy dogs were used for the intervention and commissioned through the Students Services Amenities Fee (SSAF) funding. The therapy dog sessions ran from 11.30 am to 1 pm during the first two weeks of the semester examination period on four university campuses. There were announcements on the university’s main Facebook page during the examination period notifying students of the days, campus, location, and time the therapy dogs sessions were programmed. For each day, a campus was selected for the therapy dog sessions based on the high number of students attending exams. Therapy dogs and their handlers were placed in areas that many students would pass by, such as outside of the campus library and in group seating areas of the campus, where students were able to interact with the dogs. After approximately 30 s of interacting with the therapy dog(s), students were approached to confirm that they were a student enrolled in the university and to ask for their consent in participating in the study. Participants completed the Positive and Negative Affect Schedule (PANAS) survey on an iPad. Consent was obtained in response to the first question of the survey.

To be eligible to participate in the study within the intervention group, participants needed to be (i) students at the university and (ii) observed to have interacted with the therapy dogs for at least 30 s. Students who did not provide consent were still able to interact with the dogs but were identified wearing a dot sticker to prevent them from being involved in the research data collection.

### 2.3. Control Group

During the same day, participants in the control group also completed a PANAS survey. Students on campus, such as outside the campus library or at the food hub, were asked if they would like to participate by completing the survey on the iPad. Consent was provided in response to the first question of the survey. The control group completed a similar survey to the intervention group.

To be eligible to participate in the study within the control group, participants needed to be (i) students studying at the university, present on the campus, consent their participation, and not have interacted with the dogs on campus program. To distinguish true controls, a question was included as the second-last question in the survey to clarify whether the participant had seen the therapy dogs on campus and to determine if they may be impacted by the intervention. 

### 2.4. Follow up Groups

Both the intervention group and the control group had the option to consent to a follow-up survey in four weeks’ time. Separate emails for the two groups were sent to participants who provided consent. True controls were determined with the inclusion of a question asking if the participant had interacted with the therapy dogs on campus during the examination period. 

### 2.5. Data Collection

#### 2.5.1. Intervention and Control Group Surveys

The intervention and control groups completed a survey, including the PANAS, a 20-item self-report instrument measuring positive and negative affect [26]. A total of 10 items of the PANAS represent negative affect and another 10 represent positive affect self-rated on a Likert scale of 1–5 from 1—“very slightly or not at all”—through to 5—“extremely” based on the present moment [26]. The instrument was used in the study to measure the mood and affect of students engaging with the dogs. The PANAS has been reported to have good psychometric properties, such as construct validity and test-retest correlations [27]. The positive and negative items of the PANAS demonstrated good internal consistency, α = 84 and α = 0.88, respectively [27]. 

The survey also included various questions to classify the cohort of the participants, demographic questions about dog ownership and opinions on funding for therapy dogs on campus were included. 

Additional open-ended questions developed by the authors based on the literature were added to the survey for the intervention group to understand the student participants’ perception of dogs and their opinions on student funding being used for therapy dog programs, as below:(i)Why did you approach the dogs?(ii)Select an option for this statement: I approve my student university funding being used on a project such as PAWS. (Options: strongly disagree, disagree, neither agree nor disagree, agree, strongly agree). Why?

#### 2.5.2. Follow up Group Survey

The surveys for both the therapy dog group and control group included a question regarding whether the participant would like to consent to a four-week follow-up survey. The participants who provided an email and consented to a follow-up survey were sent an email containing another survey sent four weeks after the sessions to examine if there have been changes to their positive and negative affect. The therapy dog group and control group were sent the same follow-up survey. The groups were distinguished by the inclusion of the question about whether they had interacted with the therapy dogs on campus as the last question of the survey. The survey also consisted of the PANAS as a measure for changes in participants’ positive and negative affect four weeks after the time of survey completion. Having post-intervention and follow-up scores would allow to compare whether the intervention group had a statistically significant difference in PANAS scores four weeks after intervention for each cohort sample [28].

Other survey questions included were based on the participants’ health seeking behaviors, such as: (i)If they sought health services in the past four weeks;(ii)If they plan to seek health services in the near future.

### 2.6. Data Analysis

The data collected through the surveys was exported from Qualtrics into Microsoft Excel. The data were checked for incompletion and true controls. Participants who did not provide their course name were emailed to complete the data. Course names were coded by groups as categorized by the university. The data set was coded and inserted into the IBM Statistical Package for the Social Sciences (SPSS) (Armonk, NY, USA). The positive items of the PANAS were labeled ascendingly, while the negative items were reverse-scored to enable comparisons of the combined negative and positive affect items [29]. Total scores of the PANAS were calculated for the participants, and analyses for group differences were performed in SPSS. 

Data regarding reasons for students approaching dogs and their perceptions of SSAF funding for therapy dog programs were analyzed using Braun and Clarke’s [30] thematic analysis methods. Written data was deduced into themes to obtain participants’ perceptions, which would enhance the meaning of the data collected within the mixed methodologies [31,32]. 

### 2.7. Ethical Considerations 

The project received ethical clearance from the university ethics committee. The therapy dogs involved in the project had undergone temperament assessments and received clearance from the university’s security.

## 3. Results

### 3.1. Descriptive Statistics and PANAS Total Scores

The data set was coded and inserted into IBM SPSS. The positive items of the PANAS were retained in their original order, while the negative items were reverse-scored to enable comparisons of the combined negative and positive affect items [29]. Total scores on the PANAS were calculated for the participants. The average PANAS total scores were analyzed based on the descriptive nature of the dataset and are summarized in Table 1. Based on the average PANAS scores compared by the participants’ independent factors, there is an impact on the PANAS total scores based on these factors. 

The intervention group had a higher average PANAS total score in the 16–24 (mean = 77.8, SD = 10.6, 95% CI: 76.1–79.6) and 34–45 age groups (mean = 83.2, SD = 11.3, 95% CI: 74.6–91.9), whereas the 25–34 age group had a higher average PANAS score in the control group (mean = 73.0, SD = 12.6, 95% CI: 66.5–79.5). 

The intervention group had a higher average PANAS total score in both the male (mean = 77.1, SD = 12.6, 95% CI: 73.3–81.0) and female (mean = 78.1, SD = 10.1, 95% CI: 76.4–80.0) factors compared to the control group. 

The intervention group had a higher average PANAS total score in the “domestic” enrolled participants (mean = 77.8, SD = 11.1, 95% CI: 76.0–79.6) whereas the “international” enrolled participants had a higher average PANAS total score in the control group (mean = 70.5, SD = 14.3, 95% CI: 47.8–93.3).

The intervention group had a higher average PANAS total score among the “undergraduate” participants (mean = 77.9, SD = 10.9, 95% CI: 76.2–79.6) whereas the “postgraduate” participants (mean = 75.6, SD = 16.4, 95% CI: 60.4–90.7) had a higher average PANAS total score for the control group.

The intervention group had a higher average PANAS total score for participants in the first (mean = 77.4, SD = 10.5, 95% CI: 75.0–79.8), second (mean = 75.6, SD = 11.9, 95% CI: 71.6–79.6), third (mean = 77.8, SD = 10.7, 95% CI: 74.0–81.5) and fourth year of study than in the control group (mean = 83.8, SD = 9.6, 95% CI: 79.2–88.5).

The intervention group had a higher average PANAS total score on all campuses compared to the control group, as shown in Table 1.

The intervention group had a higher average PANAS total score in the “Business, computing, engineering, mathematics, law” (mean = 78.8, SD = 11.9, 95% CI: 75.8–81.9) and “Nursing, social sciences, psychology, medicine, health, science” (mean = 76.9, SD = 10.9, 95% CI: 74.5–79.2) course groups, whereas the control group had a higher average PANAS total score for the “Education, humanities, communication arts” course group (mean = 80.0, SD = 21.4, 95% CI: 45.9–114.1).

### 3.2. Comparing the Average PANAS Total Scores for the Study Groups

The descriptive characteristics of the study groups were analyzed and summarized in Table 2. An independent samples *t* test was used to compare the average PANAS total scores by participants in the intervention condition (n = 170) to the average PANAS total scores in the control condition (n = 95), as shown in Table 3. The *t* test was statistically significant, with the intervention group (mean = 77.6, SD = 11.0) reporting a mean difference of 8.2 (95% CI: 5.2–11.2, *p* = 0.000) with a t score of 5.385 compared to the control group (mean = 69.4, SD = 13.4).

An independent samples *t* test was also used to compare the average PANAS total scores of the follow-up groups with no significance, as shown in Table 3.

A two-tailed, paired samples *t* test was used to compare the average PANAS total scores for the intervention group (mean = 77.6, SD = 11.0) and follow up intervention group (mean = 69.3, SD = 12.0), as shown in Table 4. On average, there is a mean difference of 11.6 (95% CI: 3.2–20.0). The difference was statistically significant *p* = 0.01. 

A two-tailed paired samples *t* test was also used to compare the average PANAS total scores for the control group (mean = 69.4, SD = 13.4) to the follow-up control group (mean = 72.1, SD = 11.5), as shown in Table 4. On average, there was a difference of −13.7 (95% CI: −25.8–−1.6). The difference was statistically significant *p* = 0.03.

### 3.3. Analysis of Variance 

A factorial analysis of variance (ANOVA) was used to investigate the impact of the independent factors on the PANAS total scores of the participants. 

The ANOVA was statistically significant for factors “study group”, “year of study”, and “surveyed campus” (*p* = 0.000, *p* = 0.012, and *p* = 0.010, respectively) and suggests there is an association between the PANAS total scores and these independent factors, as shown in Table 5.

To evaluate the impact of the study group combined with the independent factors—age, gender, student type, enrolment type, year of study, surveyed campus, and course groups—a factorial ANOVA was used. When combined with the “study group” factor, age, gender, the surveyed campus, and course groups were statistically significant *p* < 0.05, as shown in Table 5. This suggests that the PANAS total scores has an association with these independent factors in combination with the “study group” factor. The therapy dogs had a large impact on the 16–24 and 35–44 age groups. Although both the female and male factors benefited from the therapy dogs, the females had a bigger impact. For the “surveyed campus” factor, Campbelltown and Parramatta South are seen to have a large impact from the therapy dogs compared to the other two campuses. The course group “Nursing, social sciences, psychology, medicine, health, science” was seen to have a greater impact from the therapy dogs compared to the other course groups.

### 3.4. Further Responses from the Surveys

Further responses from the surveys were obtained to gather the context of the participants and examine their perceptions of the therapy dog program. The majority of the participants were on campus for their exams (n = 167, 63%) compared to studying (n = 46, 17.4%), having heard about the dogs (n = 24, 9.1%), with 28% of participants selecting other and specified reasons such as a post-placement workshop and research. Within the intervention group, the majority of the participants owned a dog (n = 96, 56.5%). Within the intervention group, the majority of the participants selected “strongly agree” to the statement “being with dogs calms me down” (n = 123, 72.4%). Additionally, within the intervention group, the majority of the participants selected “strongly agree” with the statement “I approve my Student University Funding being used on a project such as PAWS” (n = 104, 61.2%). Twenty-six participants responded to the follow-up survey. A total of 2 participants utilized health services in the past 4 weeks in the follow-up intervention group (13.3%) compared to one participant in the follow-up control group (9.1%). In both follow up groups, more participants selected “no” for having plans to utilize health services in the future.

### 3.5. Themes Regarding Therapy Dogs on Campus

A text box was provided in the questions “why did you choose to interact with the dogs” and to provide a reason to explain the option participants selected for “I approve my Student Services and Amenities Fee (SSAF) Funding being used on a project such as PAWS” within the intervention group survey. Thematic analysis was used to deduce the responses into themes. Participants’ responses may have been divided into more than one theme. 

Predominant themes that emerged included:(1)Positive reaction to dogs

A significant number of participants (n = 141) responded positively to the therapy dogs. Participants stated that they missed their own dog at home or had previously owned a dog. Many participants made positive comments about the characteristics of dogs. For example, one participant stated “cause they are calm and cute”. 

(2)Dogs having an effect on their own feelings and psychological wellbeing

Forty-eight participants implied that dogs have an effect on their feelings and psychological wellbeing. Seven participants mentioned that it made them feel better for exams and helped with their stress. For example, one participant stated “to help me relax and focus my attention onto something else before my exam this afternoon”.

(3)Positive reaction to the therapy dog program

Twenty-one participants responded positively to the therapy dog program. Some participants mentioned that the program is a good idea and has benefits for the students. For example, one participant stated, “these sorts of initiatives directly impact students in a practical way”.

(4)The impact on university experience/environment

Fifteen participants stated that the dogs improve the feel of the university environment. For example, one participant stated, “dogs create a positive environment for learning and create a calm and happy environment”.

## 4. Discussion

The effectiveness of therapy dog interventions for university students was investigated in this study. It was found that exposure to therapy dogs showed on average a higher mood and affect, and predominantly positive qualitative responses were obtained, as is evidenced by “Don’t have my own but I love dogs. I have anxiety and [dogs] reduces this”.

Participants in the intervention group were voluntary and self-selected their participation. The control group did not interact with or view the animals and acted as a comparison group [24], which aims to identify whether the therapy dogs were associated with the PANAS difference. The current study found that participants who interacted with the dogs had a significantly better mood and affect score compared to those who were not exposed to the dogs. This is supported by the previous literature where therapy dogs increased the mood and affect of students studying at university [22,33]. This has been supported by previous studies where there has been a reduction in stress [11,22], a reduction in negative mood, and an increase in positive mood [33]. Furthermore, these findings are in line with other studies that found that psychological stress was reduced after interacting with the therapy dogs [1,19]. The current study contributes to the existing information regarding the use of therapy dogs for university students, with a larger sample size than most previous studies.

In the current study, follow-up groups were included to investigate the post-effects of the therapy dog programs. However, with the low response rate, it is difficult to make statements about the lasting effects of exposure to the therapy dogs. There was not a statistically significant mean difference between the follow-up groups, and there is limited information on how the intervention affected participants’ health-seeking behaviors. Studies using follow-up data and associating this with health-seeking behaviors require further investigation. 

The influence of independent factors on PANAS total scores for the effectiveness of therapy dogs was investigated. Studies have found that therapy dog interventions may have more benefit for female students than male students [34]. This is supported by the findings of the study, as females in the intervention group were seen to have a higher average PANAS total score than the males. The study explored the influence of other independent factors, as it is important to consider these influences on the effectiveness of the therapy dog interventions [34]. No other studies to date have reported findings on these other influences, and this study adds new information to the investigation of factors associated with the effectiveness of therapy dog interventions. It was found that, on their own, the factors “study group”, “year of study”, and “campus of survey” were statistically significant and had an association with the positive PANAS total scores of the participants. When combined with “study group”, all factors of “age”, “gender”, “surveyed campus”, and “course groups” had a significant influence on the PANAS total scores of participants. Thus, independent factors of the participants are important to consider and may play a role in affecting the effectiveness of the therapy dog interventions.

It was found that the 16–24 and 35–44 age groups had a higher PANAS score than the 25–34 age group. It is difficult to ascertain why this may be. In Australia, pet ownership is highest for those aged 18–24 years (70%) followed by those aged 40–54 years (66%) [35], and this may help explain a pre-existing familiarity with and appreciation of the calming effect of dogs in these age groups. Women are also the group with the highest proportion of pet owners [36], and this may help explain the more dominant affectionate connection to and calming effects on this gender. 

Thematic analysis was conducted to understand the reactions to the therapy dogs and the use of university funding from the participants in the intervention group. The responses obtained from the participants support the literature on students’ perceptions of therapy dogs on campus. As in the previous literature, therapy dogs were found in this study to create calm and happy feelings [2,15]. Shared by one participant, where the dogs do not require anything extra from participants “Because they are so friendly and they don’t want anything from you except company”. Similarly, for studies conducted in the university environment, student responses indicated that the therapy dogs did promote a relaxed and comfortable environment [9,20]. This was supported by participants statements “just had a bad exam and this would make me feel a little better” and “Just had a bad exam and this would make me feel a little better”. Furthermore, the majority of the responses were positive comments regarding the therapy dogs and the program [1], supported by many statements of “I love dogs” This supports the quantitative findings suggesting that having therapy dogs on campus can improve mood and psychological wellbeing during examination time.

The study highlighted how a relatively simple health promotion intervention using dogs created an environment that supported student health and well-being. Such health promotion interventions are good examples of how educational institutions can fulfill their social responsibility by promoting the health and wellbeing of students and staff in addition to their core businesses of education and research generation [36]. Additional benefits may include improved student retention and progression, given the program may have also initiated student connection and community, a known contributor to reducing attrition [37]. Future studies could also investigate these longer-term benefits for students and universities.

## 5. Limitations

Despite using a standardized assessment tool, it is difficult to generalize the study findings to the university population due to the sample size, the unequal distribution of the sample characteristics, and the unequal number of participants among the intervention and control groups. Sampling bias may be caused by the recruitment method using voluntary participants and the low response rate for the follow up surveys. Furthermore, it is difficult to ascertain whether the PANAS scores were directly influenced by the interaction with the therapy dogs.

## 6. Future Studies

The use of a pre-post study methodology comparing baseline and post-intervention scores would be an ideal design for future studies. However, due to challenges with the natural university location and program arrangements, the current study was unable to create an experimental environment where participants could be surveyed before interacting with the therapy dogs. It was also difficult to arrange data collection after interacting with the therapy dogs, as students wanted to go directly to their examinations. Future studies should investigate the direct effect of the exposure to therapy dogs using a confirmatory trial with pre-and-post intervention with a standardized outcome measure. Also, future studies should investigate the intricacies behind the influence of certain independent factors. The study should also be applied to other population groups.

## 7. Conclusions

The findings of this study suggest that participants exposed to therapy dogs during the final examination period had a better mood and affect than those who were not exposed. This suggests that having therapy dogs on campus may be an effective way of improving students’ mood and affect during stressful periods such as examinations. As a result, health promotion professionals in education and health settings should explore the use of therapy dogs to improve mood and affect for students, staff, and clients during exams and other times of stress within the university environment.

## Figures and Tables

**Table 1 ijerph-20-04759-t001:** Characteristics of the participants and PANAS mean scores.

Factor		Intervention Group (n = 170)		Control Group (n = 95)		Total (n = 265)	
		N (%)	Mean	N (%)	Mean	N (%)	Mean
Age							
	16–24	144 (84.7)	77.8	89 (93.6)	68.9	233 (87.9)	74.4
	25–34	17 (10)	73.0	4 (4.2)	82.5	21 (7.9)	74.8
	35–44	9 (5.3)	83.2	1 (1.1)	65.0	10 (10.5)	81.4
	45+	0	-	1 (1.1)	70.0	1 (0.4)	70.0
Gender							
	Male	43 (25.3)	77.1	52 (54.7)	73.3	95 (35.8)	75.0
	Female	125 (73.5)	78.1	42 (44.2)	64.3	167 (63.0)	74.7
	Non-identified	2 (1.2)	56.5	1 (1.1)	83.0	3 (1.1)	65.3
Student type							
	Domestic	152 (89.4)	77.8	91 (9.5)	69.4	243 (91.7)	74.6
	International	18 (10.6)	76.2	4 (4.2)	70.5	22 (8.3)	75.1
Enrolment type							
	Undergraduate	159 (93.5)	77.9	88 (92.6)	68.9	247 (93.2)	74.7
	Postgraduate	11 (6.5)	74.0	7 (7.4)	75.6	18 (6.8)	74.6
Year of study							
	First	77 (45.3)	77.4	44 (46.3)	65.3	121 (45.7)	73.0
	Second	37 (21.8)	75.6	21 (22.1)	71.7	58 (21.9)	74.2
	Third	34 (20)	77.8	19 (20)	71.6	53 (20)	75.6
	Fourth	19 (11.2)	83.8	11 (11.6)	77.7	30 (11.3)	81.6
	Other	3 (1.8)	67.7	0	-	3 (1.1)	67.7
Surveyed campus							
	Campbelltown	45 (26.5)	77.3	40 (42.1)	64.1	100 (37.7)	71.1
	Hawkesbury	14 (8.2)	77.8	1 (1.1)	73.0	15 (5.7)	77.5
	Parramatta South	80 (47)	77.9	17 (17.9)	64.9	105 (39.6)	75.6
	Penrith	31 (18.2)	77.4	37 (38.9)	77.2	71 (26.8)	77.3
Course groups							
	Business, computing, engineering, mathematics, law	52 (30.6)	78.8	37 (38.9)	73.7	89 (33.6)	76.7
	Education, humanities, communication arts	18 (10.6)	78.3	4 (4.2)	80.0	22 (8.3)	78.6
	Nursing, social sciences, psychology, medicine, health, science	84 (49.4)	76.9	44 (46.3)	65.1	128 (48.3)	72.8

Note: Survey items were ordered as per original format. Items: “distressed”, “upset”, “guilty”, “scared”, “hostile”, “irritable”, “ashamed”, “nervous”, “jittery”, and “afraid” were reverse scored to calculate the total PANAS score.

**Table 2 ijerph-20-04759-t002:** Differences between study groups.

	N	Mean	Std. Deviation ^1^	Minimum	Maximum	95 % CI ^2^	
						Lower Bound	Upper Bound
Intervention	170	77.6	11.0	42	99	76.0	79.3
Control	95	69.4	13.4	37	96	66.7	72.2
Follow up intervention	15	69.3	12.0	40	85	62.6	75.9
Follow up control	11	72.1	11.5	58	98	64.4	79.8

1. Confidence interval. 2. Standard deviation.

**Table 3 ijerph-20-04759-t003:** Differences between groups—independent samples test.

	Mean Difference	Std. Error Difference ^1^	95% CI of the Difference		*t*	df	Sig. (2-Tailed) ^2^
			Lower	Upper			
Intervention–control	8.2	1.5	5.2	11.2	5.4	263	0.000
Follow up intervention–follow up control	−2.8	4.7	−12.5	6.9	−0.6	24	0.553

1. Standard error difference. 2. Significance (2-tailed).

**Table 4 ijerph-20-04759-t004:** Paired samples test.

		Mean Difference	Std. Deviation ^1^	Std. Error Mean ^2^	95% CI of the Difference		*t*	df	Sig. (2-Tailed) ^3^
					Lower	Upper			
Pair 1	Intervention group–follow up intervention	11.6	15.1	3.9	3.2	20.0	3.0	14	0.01
Pair 2	Control group–follow up control	−13.7	18.0	5.4	−25.8	−1.6	−2.5	10	0.03

1. Standard deviation. 2. Standard error mean. 3. Significance (2-tailed).

**Table 5 ijerph-20-04759-t005:** ANOVA results.

PANAS Total Scores and:	Sum of Squares	df	Mean Square	F	Sig.
Study group	4116.7	1	4116.7	28.995	0.000
Age group	491.7	3	163.9	1.044	0.374
Gender	273.9	2	137.0	0.871	0.420
Student type	4.9	1	4.	0.31	0.860
Enrolment type	0.1	1	0.1	0.001	0.980
Year of study	1984.6	4	496.1	3.268	0.012
Surveyed campus	1761.7	3	587.2	3.861	0.010
Course groups	1527.7	6	254.6	1.645	0.135
Age group * study group	1136.0	2	568.0	4.075	0.018
Gender * study group	2222.8	2	1111.4	8.350	0.000
Student type * study group	23.8	1	23.8	0.167	0.683
Enrolment * study group type	441.1	1	441.1	3.124	0.078
Year of study * study group	745.8	3	248.6	1.841	0.140
Surveyed campus * study group	2821.8	7	403.1	3.086	0.004
Course groups * study group	884.9	2	442.4	3.328	0.038

Note: Follow-up groups not included in the ANOVA.

## Data Availability

Data Availability Statements are available from the authors.

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
