# Peer review of "The Impact of Campus-Based Therapy Dogs on the Mood and Affect of University Students"

_ijerph, 2023, doi:10.3390/ijerph20064759_

Round 1

Author Response

Thank you for your suggestions these have been addressed in the document upload.

Reviewer 2 Report

In the last line of your abstract, you may want to include a recommendation for colleges and/or how your research fills a gap in the research

Lines 22 and 23 make an overall negative statement about all college students that may not be true.

Line 40-provide some examples

Line 56-what is “these” referring to?

Line 60-instead of “used” maybe “possibly be part of the intervention”

Line 66-a goal directed intervention  

Lines 68 and 69 can be deleted

You may want to reorganize the order of your lit. review talking about the benefits of therapy animals and then making the connection to college students and then talking about students directly involving college students and therapy animals leading to your research question.

Line 177-Move your research questions up to the end of your lit review right before your methods section.

In the Methods section you say how you collect qualitative and quantitative information, but you do not say how.. a survey? Give more details.  You may want to combine some of the information from the sampling recruitment section.

In the Include information about the college..private/public/four year/two year/urban/rural, etc.

How many students? Age? Gender? Other demographics?

How were the open-ended questions developed?

Do not need the ANOVA figures

In your discussion, instead of just summarizing findings, you also want to say how what you found can impact universities. What do you recommend colleges do with this information? Share your ideas and voice here.

*Need a limitations section

*Need a future research section

Author Response

We thank you for your suggested changes and have addressed these in the document attached.

Round 2

Reviewer 2 Report

Possibly include subheadings in the literature review to make it more organized for the reader.

Missing some punctuation throughout. 

Author Response

Thank You for your comments, these have been addressed and actioned as below.

Headings have now been added to the literature review

The document has been reviewed and changes in punctuation have been made.